# Biomaterials and Clinical Applications of Customized Healing Abutment—A Narrative Review

**DOI:** 10.3390/jfb13040291

**Published:** 2022-12-10

**Authors:** Parima Chokaree, Pongsakorn Poovarodom, Pisaisit Chaijareenont, Apichai Yavirach, Pimduen Rungsiyakull

**Affiliations:** Department of Prosthodontics, Faculty of Dentistry, Chiang Mai University, Chiang Mai 50200, Thailand

**Keywords:** customized healing abutment, implant abutment, dental abutment, PEEK, PMMA, titanium, zirconia, resin composite

## Abstract

Customized healing abutments have been introduced in clinical practice along with implant surgery to preserve or create natural-appearing hard and soft tissue around the implant. This provides the benefits of reducing the overall treatment time by eliminating the second stage and reducing the elapsed time of the fabrication of the final prostheses. This article aims to review the types and properties of materials used for the fabrication of customized healing abutments and their clinical applications. Articles published in English on customized healing abutments were searched in Google Scholar, PubMed/MEDLINE, ScienceDirect, and the Scopus databases up to August 2022. The relevant articles were selected and included in this literature review. Customized healing abutments can be fabricated from materials available for dental implants, including PEEK, PMMA, zirconia, resin composite, and titanium. All the materials can be used following both immediate and delayed implant placement. Each material provides different mechanical and biological properties that influence the peri-implant tissues. In conclusion, the studies have demonstrated promising outcomes for all the materials. However, further investigation comparing the effects of each material on peri-implant soft and hard tissues is required.

## 1. Introduction

Implant treatment nowadays has become a common modality. Several factors have been reported, related to implant success; one of them is the development and maintenance of healthy peri-implant soft tissue [1]. The traditional method involves implant placement with a submerged protocol, followed by second-stage surgery after the osseointegration period. This increases the complications from additional surgery. To avoid stage-two surgery, a flapless, non-submerged protocol was proposed with either a one-piece implant or a two-piece implant with the immediate connection of a transmucosal healing abutment [2]. A two-piece implant with a transmucosal smooth hyperbolic neck present platform-switch, and a smooth transmucosal neck protruding through the peri-implant soft tissue, was shown to reduce marginal bone loss in a 3-year prospective cohort study [2]. However, with this technique, the transmucosal contour could not be altered after the implant placement. Another method to avoid stage-two surgery and create peri-implant soft tissue is with an immediate connection of a transmucosal healing abutment. In the conventional method, a standard healing abutment is usually connected to the implant fixture during the second surgery. Based on the round circular shape of a standard healing abutment, the result is a round, unnatural-looking soft tissue profile [3]. Additional appointments might be required for further tissue conditioning via multiple gradual adjustments of the provisional restoration; otherwise, difficulties upon insertion of the final prostheses could lead to the patients’ discomfort or mechanical complications, such as screw-loosening due to the rebound force from the compressed peri-implant tissue [4]. Multiple disconnections and re-insertions of the provisional restoration can potentially compromise the healing process [4,5]. Therefore, some clinicians have suggested the utilization of customized healing abutments to provide a better emergence profile of the peri-implant tissues. A customized healing abutment is designed by modifying the size and transmucosal shape of the healing abutment to mimic the natural profile of an emerging tooth. Then, it is connected to the implant on the day of surgery and left undisturbed until osseointegration and tissue maturation are achieved. Customized healing abutments can be fabricated with different materials and techniques, depending on their clinical applications. The variations in the properties of dental materials nowadays, therefore, produce different effects on the peri-implant tissues. This narrative review aims to revise the materials used for customized healing abutments with their properties related to peri-implant tissue maturation and their clinical applications. A literature search of electronic databases was conducted using Google Scholar, PubMed/MEDLINE, ScienceDirect, and the Scopus databases up to August 2022. The search keywords including combinations of terms such as “customized healing abutment”, “custom healing abutment”, and “custom abutment” were used to search and obtain data about the utilization of customized healing abutments. Experimental studies, both in vitro and in vivo, case reports, and peer-reviewed articles published in English were included. Then, these articles were reviewed, and the authors classified the materials being used for customized healing abutments including PEEK, PMMA, zirconia, resin composite, and titanium. Another literature search was conducted using the same databases for each previously mentioned material with search keywords such as “properties”, “biocompatibility”, and “peri-implant tissue response”. Articles in English, including experimental studies, case reports, and review articles, were included. Non-English publications were excluded from this review.

## 2. Customized Healing Abutment

An implant healing abutment serves two roles in dental implant treatment. The first is to promote the healing of the peri-implant soft and hard tissues during the healing phase, including the initiation of soft-tissue contouring. The second is to protect the implant site during the initial post-surgical healing stage from the accumulation of plaque or debris [6]. A healing abutment can be classified as a standard or customized healing abutment. A standard healing abutment from a manufacturer is usually prefabricated in a cylindrical, non-hex shape to allow ease of insertion in any direction [7]. The connection and tissue maturation result in a round peri-implant gingival emergence profile, which requires further gingival conditioning to shape the tissue into the desired form, unless there are difficulties upon the final prosthesis delivery, or mechanical complications such as abutment screw loosening [8,9]. In the esthetic zone, including the anterior and premolar teeth, the use of provisional restoration with a dynamic compression technique is widely used in both immediate and delayed placement [10,11,12,13,14,15]. However, immediate provisionalization must be restricted in cases where an occlusal load cannot be avoided. Immediate provisionalization, therefore, is rarely used in the posterior region, where occlusal forces may not be strictly eliminated. In such cases, a standard prefabricated healing abutment is connected and allowed peri-implant soft tissue to heal until sufficient osseointegration is achieved, which leads to a prolonged overall treatment duration. The idea of modifying the contour of the standard healing abutment was first described by Pow and McMillan [8]. With their technique, a standard healing abutment was modified by creating retentive grooves on its surface, followed by adding auto-polymerized poly (methyl methacrylate) (PMMA) resin to create a natural gingival profile for the final restoration, without the need for provisional restoration. A natural soft tissue profile was observed two weeks after the insertion. The authors also mentioned that, with the use of this modified healing abutment, the custom abutment and final prosthesis can be easily delivered without soft tissue entrapment, and only minor discomfort without the use of local anesthesia was reported. The use of a custom-shaped healing abutment provides a proper soft tissue contour at the time of implant placement and also lacks occlusal contact (Figure 1).

Customized healing abutments can be fabricated from various materials commonly used in dentistry. Different types of materials contribute different properties. Since customized healing abutments are usually connected following implant placement, the differences in properties from each material can influence the peri-implant tissue healing process, as well as the tissue maturation [16].

## 3. Materials Used for Customized Healing Abutment and Their Properties

### 3.1. Materials Used for Customized Healing Abutment

The materials used for the fabrication of customized healing abutments are those commonly used in dentistry, including: Polyetheretherketone (PEEK) [17,18], Polymethyl methacrylate (PMMA) [19,20,21,22,23,24,25,26,27,28], zirconia [7], titanium [29], and resin composite [30]. They can be fabricated from monolithic materials or in combinations of these materials.

#### 3.1.1. Polyetheretherketone (PEEK)

PEEK is a synthetic, tooth-colored thermoplastic polymer which belongs to the PAEK (polyaryletherketone) family [31]. PEEK presents superior physical, mechanical, and biological properties for biomedical applications, such as in orthopedics and dentistry [32]. The structure of PEEK contains repeated aromatic rings of the ether groups, which provide structural flexibility, and repeated ketone groups that provide rigidity [16]. PEEK gained popularity in dentistry as a substitution material in patients allergic to metal. PEEK can be used as a metal-free framework material for fixed and removable prostheses and various components in implant dentistry including implant fixtures, implant abutments, provisional abutments, and healing abutments; other applications include endocrowns and occlusal splints [33]. Although PEEK presents superior properties, the fabrication of customized PEEK healing abutments requires the utilization of CAD/CAM technology, thus limiting the chairside fabrication of pure PEEK customized healing abutments. Virtual designs using dental implant software allow monolithic PEEK customized healing abutments to be fabricated prior to implant surgery and be inserted after the implant placement [34]. As the chairside fabrication proceeds, a temporary PEEK cylinder can be used with a flowable composite to capture the outline of the tooth socket [35]. When customized healing abutments are fabricated from PEEK, they can be adjusted to fit the implant site by adding or reducing the contour intraorally [36]. When combined with the composite, the roughening of the PEEK surfaces increases the bond strength with the veneering resin [37].

#### 3.1.2. Polymethyl methacrylate (PMMA)

Poly(methyl methacrylate) (PMMA) is the most commonly used polymer in dentistry. In general, PMMA polymer is prepared using a liquid methyl methacrylate (MMA) monomer along with cross-linking agents and inhibitors, and a pre-polymerized PMMA powder together with additives such as pigments and nylon or acrylic synthetic fibers [38,39]. The PMMA polymerization reaction occurs by the free radical addition and polymerization of methyl methacrylate (C_5_O_2_H_8_) to poly methyl methacrylate (C_5_O_2_H_8_)_n_ [40]. Auto-polymerization or self-cured PMMA, is widely used in direct provisional restorations because of several advantages, including low cost, acceptable aesthetic qualities, good wear resistance, high polishability, color stability, and a good marginal fit with optimal transverse strength. Self-cured PMMA is a well-reported residual MMA monomer, which has the possibility to cause irritation in some patients [16,41]. The salivary environment can lead to the degradation of PMMA by increasing the diffusion of residual MMA monomer due to the polar properties from the immersed resin molecule [42]. Dimensional contraction and exothermic reaction during polymerization could also interfere with the oral status. Self-cured PMMA may present an ease of chairside manipulation but may limit the fabrication of customized healing abutments due to its inferior properties, including water degradation, low wear-resistance, and low fracture-resistance, which lead to possible crack formation and material fracture if an occlusal load is applied [40]. Recently, the benefit of CAD/CAM technology has allowed PMMA manufacture via rapid prototyping and milling techniques [43,44]. CAD/CAM PMMA demonstrated better mechanical properties compared to conventional heat-cured and self-cured PMMA [44,45,46,47]. Several articles reported the utilization of CAD/CAM PMMA as a customized healing abutment for socket closure following immediate and delayed implant placement [19,20,21,22,23,24,25,26,27,28].

#### 3.1.3. Zirconia

Zirconia is a crystalline dioxide of zirconium which provides optimum properties in dentistry, including superior toughness, strength, fatigue resistance, excellent wear properties, and biocompatibility [48,49,50]. Dental zirconia generally refers to a modified yttria tetragonal zirconia polycrystal (Y-TZP). Yttria is added to stabilize the crystal structure transformation during firing at an elevated temperature, and to improve the physical properties of zirconia. The zirconia tetragonal-to-monoclinic phase transformation after exposure to stress is known as transformation toughening. During this zirconia phase transformation, the unit cell of monoclinic configuration occupies about 4% more volume than the tetragonal configuration, which is a relatively large volume change. This inhibits crack propagation, which lessens fractures and the failure of materials when used as customized healing abutments [51]. Zirconia can be used in fixed prostheses and also in implant components [52]. When zirconia was fabricated with CAD/CAM technology, the excessive machining, which can lead to tensile stresses on the material surface, was mentioned and may cause a direct impact on the material properties [53]. When zirconia customized healing abutments were fabricated, it was recommended to minimize the further adjustment of these abutments to reduce the propagation of this phenomenon [7]. Although zirconia provides several good properties, zirconia for customized healing abutments might not be popular because of its higher cost compared to other materials [7].

#### 3.1.4. Resin Composite

Resin composite is one of the most commonly used dental materials for direct restoration. Dental resin composite typically comprises a mixture of dental resins and diverse inorganic fillers. Resins contain two or more monomers to achieve the desired mechanical properties [54]. The base monomers include bisphenol A glycidyl methacrylate (Bis-GMA), ethoxylated bisphenol A dimethacrylate (Bis-EMA), urethane dimethacrylate (UDMA), and cross-linking diluents to adjust the viscosity of the mixtures; triethylene glycol dimethacrylate (TEGDMA), decanediol dimethacrylate (D3MA), and 2-hydroxyethyl methacrylate (HEMA) are the most common compositions of dental resin composites. The reactive methacrylate groups polymerize through light-initiated curing that causes a chain-reaction polymerization followed by a cross-linking reaction. These processes are associated with the properties of resin composite, including the mechanical and physical properties. The fillers include silica-based particles, glass-ceramics, ceramics, metals, mineral particles, or polymer-based particles, which are dispersed in different concentrations in order to improve their properties [54,55]. Two types of resin composites have been used for the fabrication of customized healing abutments: flowable [35,56,57], and packable materials [30]. Studies have demonstrated the application of resin composite onto other materials, such as PEEK or titanium provisional abutments, to capture the outline of the extraction socket in immediate implant placement.

When resin composite bonds with other material, the cured composite provides a clear three-dimensional representation of the peri-implant soft tissue profile, facilitating an accurate transfer to the technical laboratory. Resin composite presents a low elastic modulus but high fracture resistance and tensile strength. Both ceramic and composite resin abutments have been shown to have a similar failure rate during in vitro accelerated fatigue testing [58,59], suggesting that resin composite could be used to fabricate healing abutments.

#### 3.1.5. Titanium

Titanium is generally used in implant treatments. Commercially pure titanium (cp-Ti) and titanium alloy (Ti–6Al–4V) remain the most widely used materials for biomedical applications [60]. Titanium is used in alloys to fabricate dental implants due to its good mechanical properties, low density, and good bone-contact biocompatibility. Cp-Ti is available in four grades, numbered 1 to 4, according to the purity and the processing oxygen content [61]. The differences in composition demonstrate variations in corrosion resistance, ductility, and strength. The most widely used in dental implants is grade 4 cp-Ti due to its mechanical strength, and it contains the highest oxygen content (around 0.4%) [62]. Implant components such as fixtures, abutments, screws, and healing abutments can be fabricated with cp-Ti and its alloy due to their excellent biocompatibility, corrosion resistance, high strength, and low modulus of elasticity [63]. Customized healing abutments from titanium require the use of CAD/CAM implant software and have been reported in one study [29].

### 3.2. Properties of Material Used for Customized Healing Abutments

Regarding their purpose to create natural and optimal peri-implant tissue, the materials being used for the fabrication of customized healing abutments should provide sufficient physical and mechanical properties to maintain their function in the oral environment during the healing period. Moreover, they should demonstrate biocompatibility that facilitates soft tissue healing and maturation [64,65].

#### 3.2.1. Functional Properties

The materials for customized healing abutments should demonstrate the durability to remain functional in oral environments, as well as the dimensional stability to guide peri-implant tissue maturation. The related properties are listed as follows:Modulus of elasticity: it’s a numerical expression, indicating the measure of stiffness in a material. This allows the behavior of a material under a load to be calculated. When the material connected to the implant receives an occlusal load, one with a higher elastic modulus may deliver more stress to the cortical bone [66,67]. PEEK, CAD/CAM, and conventional PMMA and resin composites demonstrate an elastic modulus closer to human cortical bone than titanium and zirconia [68,69]. Zirconia presents the highest elastic modulus among these materials. This indicates that customized healing abutments made from titanium or zirconia can possibly cause more damage to bone than polymer and resin materials, especially when an implant is placed at the crestal level, which was shown to be significant to bone apposition at this level due to bone remodeling from the direct contact between the healing abutment and the crestal bone.Fracture resistance and flexural strength: Fracture resistance is a material property that describes the material’s capacity to resist fracture when experiencing a crack, while flexural strength is the ability of the material to withstand bending forces applied perpendicularly to its longitudinal axis. These properties are important because they contribute to resistance to occlusal loads that may cause a fracture or distortion of materials during the tissue maturation process. Zirconia and titanium present the highest strength values; PEEK has been reported to have lower fracture resistance than titanium but higher or comparable to zirconia and ceramic [60,70,71]. CAD/CAM PMMA was reported as having slightly lower fracture resistance compared to PEEK material [72]. A few minor mechanical complications were reported, such as the loosening of CAD/CAM PMMA customized healing abutments, which did not impact the outcome [22]. CAD/CAM resin composite has been reported with the lowest fracture resistance among the provisional materials in an in vitro study [73]. Resin composites usually present a low modulus of elasticity as well as low fracture toughness to protect the opposing tooth when used as a direct restoration [55,74,75], which may limit their use as a supporting material combines with a tougher material such as PEEK or titanium cylinder [30,35,57,76].Surface roughness is known to be related to the tissue response and cell adhesion [77,78] leading to soft tissue sealing [79]. Smooth abutment surfaces with a roughness value < 0.2 μm are recommended to ensure soft tissue sealing [80], and a roughness of <0.8 μm yields less bacterial colonization [81]. Polished zirconia was reported with lower surface roughness compared to polished titanium in an in vitro study [82]. Another study confirmed that the polishabilty of zirconia results in markedly low surface roughness and may contribute to its superior tissue adhesion [83]. An in vitro study showed PEEK presents lower surface roughness than a titanium abutment [84]. The study demonstrated the reduced surface roughness of PEEK after polishing [85]. CAD/CAM PMMA presents lower surface roughness compared to conventional heat-cured and light-cured PMMA [86]. Another in vitro study reported CAD/CAM PMMA demonstrated cellular behavior similar to that of lithium disilicate (current gold standard) and is, therefore, a material suitable for use as an implant provisional prosthesis. Since this material facilitates peri-implant soft tissue maturation [87] due to its low surface roughness. Resin composites also presented low surface roughness and a significantly lower value after proper polishing [88,89,90].Contact angle and wettability: Materials with a low contact angle and considered hydrophilic have demonstrated a positive correlation with plaque accumulation [91,92]. PEEK is inherently hydrophobic with a high contact angle, thus making it bioinert [93]. CAD/CAM PMMA presents more contact angle and hydrophobicity compared to conventional heat-cured PMMA [45,46]; both materials are considered hydrophobic. Resin composites have also demonstrated hydrophobicity due to their high contact angle [94]. Titanium and zirconia present a low contact angle, thus are classified as hydrophilic [82,95,96].Thermal conductivity: Titanium presents higher thermal conductivity [97] than PEEK, PMMA, zirconia, and resin composite, thus influencing taste perception [98,99,100].

A summary of properties related to the function of customized healing abutments is presented in Table 1.

#### 3.2.2. Biological Properties

Customized healing abutments have a proper design for the final prosthesis. Therefore, the peri-implant tissue is expected to form identically to its profile. The materials suitable for customized healing abutments should therefore provide tissue response in such a way to create an accurate restorative emergence profile of the peri-implant mucosa, regenerate adequate papillae in height and width, and establish peri-implant mucosal margins in harmony with the gingival contours of the adjacent teeth [102]. The properties of the material should be biocompatible with the peri-implant mucosa and allow the healing process and soft tissue adhesion, as well as reduce biofilm and bacterial adherence [64,65].

Tissue adhesion and tissue response: Fibroblasts and epithelial cells are known to acquire the major cellular composition of the peri-implant mucosa [65]. Therefore, the effect of fibroblast and epithelial adhesion could possibly lead to promoting the peri-implant seal and tissue maturation process. Tissue adhesion is known to be associated with the material surface roughness. A smooth surface is believed to provide the adhesion of tissue. PEEK showed biocompatibility with human fibroblast cells in an in vitro study by Peng et al. [103], which showed fibroblast adhesion effectiveness, metabolic activity, and pro-inflammatory responses similar to titanium alloy incubated fibroblasts. Moreover, PEEK promoted a more prominent soft-tissue response than a titanium healing cap in an animal study [104,105]. Clinical studies on the effects of PEEK material on human peri-implant tissue are scarce. Most of the studies reported optimum peri-implant soft tissue healing after sufficient healing periods [34,36,106]. CAD/CAM PMMA has been recommended to use around peri-implant tissue more than conventional self-cured material due to having more fibroblast attachment [98,107]. Studies have reported fibroblast and epithelial adhesion and proliferation on polished zirconia surfaces and better fibroblast adhesion on zirconia than the titanium surfaces [108,109]. It is well known that epithelial cells prefer a smoother surface, and titanium shows better epithelial adhesion than fibroblasts [110]. One study demonstrated a lower surface roughness on highly polished zirconia compared to titanium [111], which may lead to better epithelial proliferation. Resin composites demonstrated higher gingival epithelial attachment compared to partially cured composites [112]. Thus, when a resin composite has been used to fabricate a customized healing abutment intraorally, it should be removed be cured extraorally to ensure complete curing and enhance the tissue adhesion. Most materials seem to provide sufficient tissue healing and tissue maturation. Studies have reported the ability to preserve the papilla and facial mucosal—as well as the soft tissue—contour of the pre-existing teeth level with PEEK and CAD/CAM PMMA customized healing abutments [113,114]. Several other studies also mentioned papilla preservation in the short-term follow-up with the use of CAD/CAM PMMA customized healing abutments [24,77,114,115]. A 1-year randomized clinical trial reported a significantly higher Papilla Index [116] in customized healing abutments than in the standard group, which indicated more papillae present at the final outcome [115]. However, another short-term clinical study reported a significant mid-facial gingival height reduction at 1 and 3-month follow-ups with customized healing abutments made from a composite on a temporary cylinder [113]. The authors mentioned a significant reduction in the lingual soft tissue margin over 6 months due to a free gingival fiber collapse after tooth extraction. Another study reported similar facial mucosal reduction in the CAD/CAM PMMA customized healing abutment group compared to the standard group, where differences in gingival phenotypes between the two groups were reported as the possible confounding factors [115].Biocompatibility: refers to the ability of a biomaterial to perform its desired function without eliciting any undesirable local or systemic reactions but generating the most appropriate beneficial cellular or tissue response [117]. The properties related to biocompatibility include corrosion resistance, which is defined as the chemical or electro-chemical reaction between a material and its environment that produces a deterioration of the material itself and its properties [118]. The literatures support the high corrosion resistance of Cp-Ti and its alloys due to the stability of the Ti oxide (TiO_2_) layer [63,119]. However, some studies reported titanium and titanium alloy are not inert to corrosive attack if the stable oxide layer is disrupted and is unable to repair [120,121]. Titanium-wear was reported at the time of the implant placement and continued under the mastication forces. The TiO_2_ nanoparticles shed on peri-implant hard and soft tissue can further lead to local irritation [122]. Zirconia presents low corrosion and thus increases biocompatibility [123]. PEEK has also demonstrated high corrosion resistance in several in vitro studies [124,125] and no monomer release [126], which causes this material to have superior biocompatibility. Another factor related to biocompatibility is the release of the material substance to oral tissues. Self-cured PMMA contributes to the residual unpolymerized monomer during the polymerization reaction and has been reported to be associated with mucosal irritation [16,41,127], as well as tissue inflammation, and cytotoxicity [41,128]. Heat-cured and CAD/CAM PMMA showed a lower residual monomer release compared to self-cured PMMA. Therefore, they are likely to provide better biocompatibility [43,98,107]. There were several studies that reported good peri-implant tissue response to CAD/CAM PMMA customized healing abutments after 1–3 months of insertion, and an ability to preserve the soft tissue architecture with slight tissue inflammation [19,20,21,22,23,24,25,26,27,28]. There were some reports of resin composite leaching substances from dental composite resins and concerns about their biocompatibility, which can affect the growth and immune responsivity of gingival fibroblasts [129,130,131]. The leaching of inorganic ions was reported as dependent on the filler composition and filler treatment [132,133]. Resin composites might compromise soft tissue healing due to their release of substances and degradation in exposure to the oral environment. Stumpel and Wadhwani suggested a method to fabricate customized healing abutments from flowable composite extraorally to minimize the uncured composite contact with the peri-implant tissue [57].Bacterial formation: refers to material susceptible to bacterial deposition, which may cause inflammation of the peri-implant tissue, thus interfering in the tissue maturation process. PEEK demonstrated equal or lower biofilm formation compared to other materials, such as zirconia or titanium, in some in vitro studies [64,134]. Moreover, in vivo studies have reported significantly less plaque accumulation on zirconia compared to titanium in the oral cavity [135,136]. Studies have concluded that bacterial adhesion was influenced by the low surface energy of zirconia. Although conventional PMMA promotes bacterial formation due to its porosity, CAD/CAM PMMA provides less bacterial accumulation due to its enhanced hydrophobicity [91,107]. Resin composites have demonstrated marked plaque accumulation, which may lead to mucosal inflammation, compared to titanium [137].

The summary of properties related to the biological responses to customized healing abutments are presented in Table 2.

## 4. Clinical Applications and Clinical Importance of Customized Healing Abutments

### 4.1. Clinical Applications

Studies have proposed the utilization of customized healing abutments in immediate and delayed implant placement with several techniques including chairside fabrication and with computer-aided technology.

#### 4.1.1. Immediate Implant Placement (IIP)

Customized healing abutments following IIP have been widely demonstrated [30,35,77,138,139,140,141,142] as having the ability to be fabricated with the direct method, which means connecting the temporary cylinder to the implant fixture and applying resin composite to capture the outline of the extraction socket intraorally [8,30,35,57,76,113,138,139,140,141,143,144], or with the indirect method. The indirect method is conducted by conventionally or digitally taking an impression of the implant position and fabricating with CAD/CAM techniques [22,23,145]. The main purposes of fabrication of customized healing abutments are the preservation of the emergence profiles of the pre-existing teeth and the closure of the implant sites [30].

#### 4.1.2. Delayed Placement

In the case of delayed implant placement, there is an absence of a pre-existing tooth. Thus, a customized healing abutment is designed to create the desired contour for the final prosthesis at the time of implant surgery. A customized healing abutment can be fabricated before the implant placement by taking an impression of the edentulous area and fabricating from a diagnostic cast, or with implant planning software and milling material [26,27,29,56]. Moreover, with advancements in dental technology, many studies have reported techniques such as using the contralateral tooth contour, where digital software was used to virtually flip the contour of the contralateral tooth to create the emergence profile of the customized healing abutment [26]. Studies have reported that the fabrication of customized healing abutments, custom impression posts, and custom abutments of the final prosthesis with an identical transmucosal contour have demonstrated favorable tissue responses [27].

### 4.2. Clinical Importance

Customized healing abutments have presented promising peri-implant tissue outcomes in several studies. With this abutment, the peri-implant tissue was guided to heal and mature without interfering with the osseointegration process [114,146]. Moreover, it could reduce the number of surgeries by eliminating a second surgery, reduce post-operative discomfort, morbidity related to the open-flap technique, and reduce the overall treatment time [115,138,145]. It also reduced the time to condition the gingiva while avoiding micromovement from an immediate load, and was shown to be minimally invasive [147]. When combined with CAD/CAM technology, a customized healing abutment can be used as the reference for the fabrication of the final prosthesis. The emergence profile can be duplicated easily and allows for a precise fabrication of the final restoration, which leads to the prevention of misfit of the abutment insertion [113]. In addition, insertion with only light contact pressure to the soft tissue could lead to long-term stability [148]. The effects of the customized healing abutments on the peri-implant mucosa are presented in Table 3.

## 5. Conclusions

The present article reviews the types of materials used for customized healing abutments, including PEEK, PMMA, zirconia, resin composite, and titanium, as well as the mechanical and biological properties of each material. According to some experimental studies and case reports, all the materials seem to provide sufficient properties to remain in the oral environment during osseointegration and provide the benefits of peri-implant soft and hard tissue preservation. However, most of the studies have reported only descriptive outcomes, such as the optimum soft tissue results. To the best of the authors’ knowledge, there is a lack of studies that compare the effects of different materials used for fabricating a customized healing abutment. Further measurement guidelines and experiments should be conducted to analyze the benefits of customized healing abutment utilization. Future studies should focus on identifying appropriate materials for customized healing abutments that provide optimal peri-implant tissue outcomes to enhance success in the implant treatment.

## Figures and Tables

**Figure 1 jfb-13-00291-f001:**
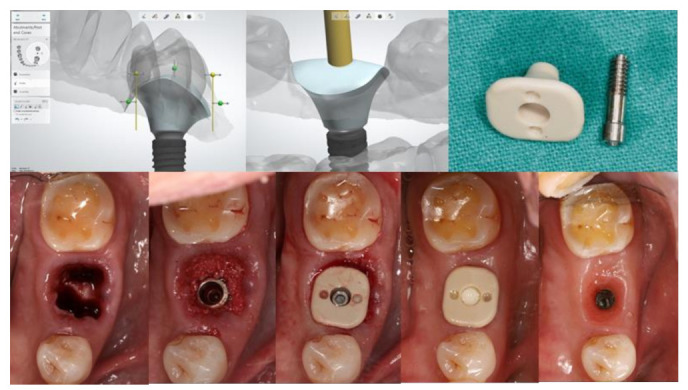
Customized healing abutment was designed with an outline of the tooth shape with transmucosal contour to preserve soft tissue before being milled with PEEK Material.

**Table 1 jfb-13-00291-t001:** Functional properties of materials used for customized healing abutments.

Properties	PEEK	PMMA	Zirconia	Titanium	Resin Composite	Reference
Elastic modulus	Low	Low	Highest	High	Low	[68,69]
Flexural strength	High	Moderate	High	High	Moderate	[55,60,69,70,71,72,73,74,75,101]
Fracture toughness	High	Self-cured; Low CAD/CAM; High	High	High	Moderate	
Surface roughness	Low	Self-cured; High CAD/CAM; Low	Lowest	Low	Low	[82,83,84,85,86,87,88,89,90]
Color	White	Tooth-colored	Tooth-colored	Greyish	Tooth-colored	[97,98,99,100]
Thermal conductivity	Low	Low	Low	High	Low	
Hydrophobicity	High	CAD/CAM; High	Low	Low	High	

**Table 2 jfb-13-00291-t002:** Biological properties of materials used for customized healing abutments.

Properties	PEEK	PMMA	Zirconia	Titanium	Resin Composite	Reference
Tissue adhesion	Good	CAD/CAM; Good	Very good	Good	Good	[34,36,104,105,106,108,109,111,112]
Biocompatibility	Very good	Self-cured; Acceptable CAD/CAM; Good	Good	Good	Acceptable	[16,41,43,63,98,107,119,123,124,125,127,129,130,131]
Bacterial formation	Low	Self-cured; High CAD/CAM; Low	Low	Low	Moderate	[64,107,134,135,136,137]

**Table 3 jfb-13-00291-t003:** Effect of customized healing abutment on peri-implant supporting tissue compared with other protocols. The change of each parameter was compared to the baseline before tooth extraction.

Effect		Cover Screw (Submerged)	Standard Healing Abutment	Customized Healing Abutment	Reference
**Tissue volume**		Decreased	-	Stable	[114]
**Soft tissue**	Horizontal contour	Comparable -	- Decreased	Comparable Stable	[114] [146]
	Vertical contour (Facial mucosal level)	- Comparable	Improved -	Decreased Comparable	[115] [114]
	Papilla level	-	Decreased	Preserved	[115]
**Hard tissue**		-	Slightly decreased	Preserved	[149]
	Horizontal contour	Decreased	-	Preserved	[145,149]
	Vertical contour (Proximal bone level)	-	Comparable	Comparable	[115]
**Esthetic and** **Patient** **satisfaction**	Pain NRS	-	Higher	Lower	[34]
	PES change	-	Slightly decreased	Stable	[115]

NRS = Numerical rating scale. PES = Pink Esthetic Score.

## Data Availability

Not applicable.

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
