# Peer review of "Biomaterials and Clinical Applications of Customized Healing Abutment—A Narrative Review"

_jfb, 2022, doi:10.3390/jfb13040291_

Round 1

Reviewer 1 Report

the manuscript appears interesting for the journal. It requires some modifications before acceptance. Here I report issues to be resolved:

Abstract should be modified.

Remove some sentences as they are redundant, such as “Relevant articles were selected and included in this literature review.”  ”Several methods 14were proposed in the fabrication of customized healing abutments as well as various types of ma- 15

terials.”

Please state clear conclusions in the abstract, use those reported in the conclusion section

 Please specify that the study is a narrative review in the text.

Also specify the search strategy. (keywords, type of papers included and excluded and reasons).

Conclusions could be summarized in key points

In the introduction, the use of one piece or two piece exposed neck should be mentioned. The reader could have interest to know that non submerged and or tissue level placement are being used to avoid second stage surgery. I recommend to include a paragraph based on recent papers that refers to the factors Affecting Soft and Hard Tissues Around Two-Piece Transmucosal Implants, regarding platform-swtich tissue level implants placed at tissue level.

Please include in pages 2-4 the limitations of each materials used as custom abutment (fabrication, degradation, wear, presence of fractures and potential failures).

Page 5 line 191-192 use “studied” instead of “studies”. Moreover, the sentence is difficult to follow. Is it possible to modify it?

Psge 5 line 209 “ it is known to be related”

Line 219 remove dot after “another”

Line 236 modify the last sentence, unclear

Page 7 regarding titanium tribological properties and wear after implant insertion, some recent reviews and histological studies could support the statement, please include some of the most recently published studies on Microanalysis at Bone-Implant Region of Retrieved Implants and regarding Titanium Wear of Dental Implants from Placement, under Loading and Maintenance Protocols.

Table 3 please use “compared” instead of compare.

Moreover, I do not understand the term reduced in the table.

Case report in a review is not necessary and is redundant. I suggest to remove it as it does not add anything to the paper.

Author Response

Point-by-Point Responses

I would greatly appreciate the constructive suggestions. For the convenience of further review, we highlighted all the corresponding revision by using “Track Change” function in the text. The point-by-point responses are provided for clarifying the issues raised, as follows.

Reviewer #1

Comment 1

The manuscript appears interesting for the journal. It requires some modifications before acceptance. Here I report issues to be resolved:

Abstract should be modified.

Remove some sentences as they are redundant, such as “Relevant articles were selected and included in this literature review.”  ”Several methods were proposed in the fabrication of customized healing abutments as well as various types of materials.”

Please state clear conclusions in the abstract, use those reported in the conclusion section

Response:

We thank the reviewer for the constructive comment. We have modified the abstract by removing the redundant sentence which is “Several methods were proposed in their fabrication of customized healing abutments as well as various types of material”. We also stated the conclusion in the abstract which included types of materials, mechanical and biological related properties and clinical applications of customized healing abutment. Please see the modified abstract on page 1.

Comment2

Please specify that the study is a narrative review in the text.

Response:

                        Thank you for your comment. We added “narrative” to our title. We also specified our study as a narrative review in the introduction section.

Comment 3

Also specify the search strategy. (keywords, type of papers included and excluded and reasons).

Response:

We appreciated your concern about the search strategy. We have added the additional paragraph specified search strategy in the introduction section on page 2. This includes the keywords we used such as “customized healing abutment”, “custom healing abutment,” and “custom abutment” for gathering data of customized healing abutment. The types of paper included also mentioned as experimental studies, case reports and peer-reviewed articles. Moreover, from the pooled data we classified materials used and included “PEEK”, “PMMA” “resin composite,” “zirconia” and “titanium” as keywords in our search strategy. For each material, we also added keywords such as “mechanical properties” and “biological properties” to obtain related studies for each material. English papers that related to our review were included. Non-related and non-English papers were excluded from our review.

Comment 4

Conclusions could be summarized in key points

Response:

We thank the reviewer for the comment. We have revised and summarized the conclusion section in key points from our article. These include types and properties of materials, clinical applications about using customized healing abutment from recent studies. Please see the adjusted conclusion section.

Comment 5

In the introduction, the use of one piece or two piece exposed neck should be mentioned. The reader could have interest to know that non submerged and or tissue level placement are being used to avoid second stage surgery. I recommend to include a paragraph based on recent papers that refers to the factors Affecting Soft and Hard Tissues Around Two-Piece Transmucosal Implants, regarding platform-swtich tissue level implants placed at tissue level.

Response:

Thank you very much for pointing out this. It would be better comprehension for reader to provide such information. We added additional paragraph mentioned about one-piece and two-piece implant neck in the first part of the introduction section along with the purpose of non-submerged protocol which is to avoid second stage surgery. We referred to the platform-switch tissue level implant and its effect according to the recent papers you had suggested. Please see the introduction section on page 1.

Comment 6
                        Please include in pages 2-4 the limitations of each materials used as custom abutment (fabrication, degradation, wear, presence of fractures and potential failures).

Response:

Thanks you very much for this concern. The limitations for each materials used as customized healing abutment were added on each material section. Please see added paragraph on page 2 to 5.

Comment 7

Page 5 line 191-192 use “studied” instead of “studies”. Moreover, the sentence is difficult to follow. Is it possible to modify it?

Response:

Thank you for your mentioned and concern on the sentence which might difficult to follow. We aimed to explained that when implant was placed at bone level, the healing abutment will contact directly to crestal bone, therefore, higher elastic modulus material could result in more stress concentrated on crestal bone. Thus, we modified the sentence to “When implant was placed at crestal level the significance is increased due to direct contact of healing abutment and the crestal bone that undergoes remodeling”. Please see the modified sentence on page 6.

Comment 8

Page 5 line 209 “it is known to be related”

Line 219 remove dot after “another”

Line 236 modify the last sentence, unclear

Response:

            Thank you for mentioning the errors. We corrected the error on line 209 and line 219 as you mentioned. For line 236 we removed “has low thermal conductivity” to make a clear sentence. Please see adjusted sentence line 287-289 on page 6.

Comment 9

Page 7 regarding titanium tribological properties and wear after implant insertion, some recent reviews and histological studies could support the statement, please include some of the most recently published studies on Microanalysis at Bone-Implant Region of Retrieved Implants and regarding Titanium Wear of Dental Implants from Placement, under Loading and Maintenance Protocols.

Response:

            We appreciated your kind suggestion.  We added paragraph related to titanium wear that refer to studies you had mentioned. Please see additional paragraph on page 9.

Comment 10
Table 3 please use “compared” instead of compare.

Moreover, I do not understand the term reduced in the table.

Response:

            Thank you very much for correcting the error. We change to “compared” on table 3. We also changed the term “reduced” to “decreased” as the table refer to measurement of peri-implant tissue contour and papilla level. The term “decreased” mean reduction in measured parameter with each method to cover implant fixture. The baseline was the level before tooth extraction and the measurement depended on each reference studies mentioned on the table. Please see table 3 for the modification.

Comment 11
                        Case report in a review is not necessary and is redundant. I suggest to remove it as it does not add anything to the paper.

Response:

We appreciate your suggestion. We removed the case report section since we agreed the unnecessary of this section on our paper.

Reviewer 2 Report

Thank You for sending article titled “Biomaterials and Clinical Applications of Customized Healing Abutment – A review “from J. Funct. Biomater. (MDPI) for review.

Comments:

The literature review on biomaterials and their clinical application is of great interest to MDPI readers. A review is interpreted appropriately. The article is interesting for the readership of the Journal. The paper would be attractive for a wide readership. Publishing this article is for the benefit of overall. It provides an advance towards the current knowledge. 

The effort of the authors to collect such a large amount of literature deserves praise.

In my opinion section 5 with “Case report” is unnecessary. Besides, the photos: Fig 1, Fig 1A, Fig 3, Fig 4 and Fig 5  are very poor and have low resolution.

Author Response

Reviewer 2

Comment 1

The literature review on biomaterials and their clinical application is of great interest to MDPI readers. A review is interpreted appropriately. The article is interesting for the readership of the Journal. The paper would be attractive for a wide readership. Publishing this article is for the benefit of overall. It provides an advance towards the current knowledge.

The effort of the authors to collect such a large amount of literature deserves praise.

In my opinion section 5 with “Case report” is unnecessary. Besides, the photos: Fig 1, Fig 1A, Fig 3, Fig 4 and Fig 5  are very poor and have low resolution.

Response:

We appreciate your feedback very much. For your opinion, we decided to remove the case report section because we agreed that it is unnecessary for our paper.  We also revised Figure 1 to make it clear for reader to visualize the concept of customized healing abutment, and the resolution of the Figure has been modified.

Reviewer 3 Report

Relevant articles, such as  materials available for dental implant, including PEEK, PMMA, zirconia, titanium and resin composite were selected and included in this literature review. The review was well origanized and well shaped. I recommend it to be accepted at the current form.

Author Response

Reviewer 3:

Comment 1

Relevant articles, such as  materials available for dental implant, including PEEK, PMMA, zirconia, titanium and resin composite were selected and included in this literature review. The review was well origanized and well shaped. I recommend it to be accepted at the current form.

Response:

The authors' team wishes to express appreciation for your feedback.

Round 2

Reviewer 1 Report

manuscript can  be accepted